# Private and Efficient Meta-Learning with Low Rank and Sparse decomposition

## Abstract

Meta-learning is critical for a variety of practical ML systems – like personalized recommendations systems – that are required to generalize to new tasks despite a small number of task-specific training points. Existing meta-learning techniques use two complementary approaches of either learning a low-dimensional representation of points for all tasks, or task-specific fine-tuning of a global model trained using all the tasks. In this work, we propose a novel meta-learning framework that combines both the techniques to enable handling of a large number of data-starved tasks. Our framework models network weights as a sum of low-rank and sparse matrices. This allows us to capture information from multiple domains together in the low-rank part while still allowing task specific personalization using the sparse part. We instantiate and study the framework in the linear setting, where the problem reduces to that of estimating the sum of a rank-$r$ and a $k$-column sparse matrix using a small number of linear measurements. We propose an alternating minimization method with hard thresholding – AMHT-LRS– to learn the low-rank and sparse part effectively and efficiently. For the realizable, Gaussian data setting, we show that AMHT-LRS indeed solves the problem efficiently with nearly optimal samples. We extend AMHT-LRS to ensure that it preserves privacy of each individual user in the dataset, while still ensuring strong generalization with nearly optimal number of samples. Finally, on multiple datasets, we demonstrate that the framework allows personalized models to obtain superior performance in the data-scarce regime.

## 1 Introduction

Typical real world settings – like multi user/enterprise personalization – have a long tail of tasks with a small amount of training data. Meta-learning addresses the problem by learning a "learner" that extracts key information/representation from a large number of training tasks, and can be applied to new tasks despite limited number of task specific training data points.

Most existing meta-learning approaches can be categorized as: 1) *Neighborhood Models*: these methods learn a global model, which is then "fine-tuned" to specific tasks (Guo et al., 2020; Howard & Ruder, 2018; Zaken et al., 2021), 2) *Representation Learning*: these methods learn a low-dimensional representation of points which can be used to train task-specific linear learners (Javed & White, 2019; Raghu et al., 2019; Lee et al., 2019; Bertinetto et al., 2018; Hu et al., 2021). In particular, task-specific fine-tuning has demonstrated exceptional results across many natural language tasks (Devlin et al., 2018; Liu et al., 2019; Yang et al., 2019; Lan et al., 2019). However, such fine-tuned models update all parameters, so each fine-tuned total parameter footprint is same as the original model. This implies that fine-tuning large models –like say a standard BERT model (Devlin et al., 2018) with about 110M parameters – for thousands or millions of tasks would be quite challenging even from storage point of view. One potential approach to handle the large number of parameters is to fine-tune only the last layer, but empirical findings suggest that such solutions can be significantly less accurate than fine-tuning the entire model (Chen et al., 2020a; Salman et al., 2020). Moreover, representation learning based approaches apply strict restrictions like each task's parameters have to be in a low-dimensional subspace which tend to affect performance in general (Sec. 3).

In this work, we propose and study the LRS framework that combines both the above mentioned complementary approaches. That is, LRS restricts the model parameters $\Theta^{(i)}$ for the $i^{\text{th}}$ task as $\Theta^{(i)} := \mathcal{U} \cdot \mathbf{W}^{(i)} + \mathbf{B}^{(i)}$, where first term denotes applying a low-dimensional linear operator on

$\mathbf{W}^{(i)}$, while $\mathbf{B}^{(i)}$ is restricted to be sparse. That is the first term is based on representing parameters of each task in a low dimensional subspace while the second term allows task-specific fine-tuning but only of a few parameters. Note that methods that only allows fine-tuning of batch-norm statistics, or of the final few layers can also be thought of as "sparse" fine-tuning but with a *fixed* set of parameters. In contrast, we allow the tunable parameters to be selected from any part of the network.

The framework allows collaboration among different tasks so as to learn accurate model despite lack of data per task. Similarly, presence of sparse part allows task-specific personalization of arbitrary but a small set of weights. Finally, the framework allows tasks/users to either contribute to a central model using privacy preserving bill-board models (see Section 2.2) or also allows flexibility where certain tasks only learns their parameters locally and do not contribute to the central model.

While the framework applies more generally, to make the exposition easy to follow, we instantiate it in the case of linear models. In particular, suppose the goal is to learn a linear model for the $i^{\text{th}}$ task that is parameterized by $\theta^{(i)}$, i.e., expected prediction for the data point $\mathbf{x} \in \mathbb{R}^d$ is given by $\langle \mathbf{x}, \theta^{(i)} \rangle$. $\theta^{(i)}$ is modeled as $\theta^{(i)} := \mathbf{U}^\star \mathbf{w}^{\star(i)} + \mathbf{b}^{\star(i)}$ for all tasks $1 \leq i \leq t$, where $\mathbf{U}^\star \in \mathbb{R}^{d \times r}$ (where $r \ll d$) is a tall orthonormal matrix that captures task representation and is shared across all the tasks.

Note that estimation of the low-rank and sparse part is similar to the robust-PCA (Netrapalli et al., 2014) that is widely studied in the structured matrix estimation literature. However, in that setting, the matrices are fully observed and the goal is to separate out the low-rank and the sparse part. In comparison, in the framework proposed in this work, we only get a few linear measurements of the underlying matrix due to which the estimation problem is significantly more challenging. We address this challenge of estimating $\mathbf{U}^\star$ along with $\mathbf{w}^{\star(i)}$, $\mathbf{b}^{\star(i)}$ (for each task) using a simple alternating minimization style iterative technique. Our method – AMHT-LRS– alternatingly estimates the global parameters $\mathbf{U}^\star$ as well as the task-specific parameters $\mathbf{w}^{\star(i)}$ and $\mathbf{b}^{\star(i)}$ independently for each task. To ensure sparsity of $\mathbf{b}$ we use an iterative hard-thresholding style estimator (Jain et al., 2014).

In general, even estimating $\mathbf{U}^\star$ is an NP-hard problem (Thekumparampil et al., 2021). One of the main contributions of the paper is a novel analysis that shows that AMHT-LRS indeed efficiently converges to the optimal solution in the realizable setting assuming the data is generated from a Gaussian distribution. Formally, consider $t$, $d$-dimensional linear regression tasks (indexed by $i \in [t]$) with $m$ samples $\{(\mathbf{x}_j^{(i)}, y_j^{(i)})\}_{j=1}^m$ being provided to each of them such that

$$\mathbf{x}_j^{(i)} \sim \mathcal{N}(\mathbf{0}, \mathbf{I}_d) \text{ and } y_j^{(i)} \mid \mathbf{x}_j^{(i)} = \langle \mathbf{x}_j^{(i)}, \mathbf{U}^\star \mathbf{w}^{\star(i)} + \mathbf{b}^{\star(i)} \rangle + z_j^{(i)} \text{ for all } i \in [t], j \in [m], \quad (1)$$

where $z_j^{(i)} \sim_{iid} \mathcal{N}(0, \sigma^2)$. Below, we state our main result informally in the noiseless setting ($\sigma = 0$):

**Theorem** (Informal, Noiseless setting). *Suppose we are given $m \cdot t$ samples from $t$ linear regression tasks of dimension $d$ as in equation 1. Goal is to learn a new regression task's parameters using $m$ samples, i.e., learn the shared rank-$r$ parameter matrix $\mathbf{U}^\star$ along with task-specific $\mathbf{w}^\star, \mathbf{b}^\star$. Then, AMHT-LRS with total $m \cdot t = \widetilde{\Omega}(kdr^4)$ samples and $m = \widetilde{\Omega}(\max(k, r^3))$ samples per task can recover all the parameters exactly and in time nearly linear in $m \cdot t$.*

That is, AMHT-LRS is able to learn the underlying model exactly as long as the total number of tasks is large enough, and per task samples scale only linearly in sparsity of $\mathbf{b}$ and cubically in the rank of $\mathbf{U}^\star$. Assuming $r, k \ll d$, additional parameter overhead per task is small, allowing efficient deployment of such models in production. Finally, using the billboard model of $(\epsilon, \delta)$ differential privacy (DP) (Jain et al., 2021; Chien et al., 2021; Kearns et al., 2014), we can extend AMHT-LRS to preserve privacy of each individual. Furthermore, for similar sample complexity as in the above theorem albeit with slightly worse dependence on $r$, we can guarantee strong generalization error up to a standard error term due to privacy.

**Summary of our Contributions:**

- We propose a theoretical framework for combining the meta-learning approaches of *representation learning* and *neighborhood model*. Our model non-trivially generalizes guarantees of Thekumparampil et al. (2021); Tripuraneni et al. (2021); Boursier et al. (2022) to the setting where the parameter matrix allows a low rank plus sparse decomposition.
- We propose an efficient method AMHT-LRS for the above problem and provide rigorous total sample complexity and per-task sample complexity bounds that are nearly optimal (Theorem 1).

- We provide a DP variant of AMHT-LRS that guarantees user level privacy. At a high-level we show that under $(\epsilon, \delta)$-DP, one can obtain a generalization error as a non-private version but with an additional error due to privacy budget (see Theorem 3).
- We demonstrate experiments on synthetic data and Movielens dataset using linear models (Sec. 3) and toy neural nets (Appendix A); apart from showing the advantage of our framework, they also show limitations of only using representation learning or a single model among other baselines.

**Technical Challenges (Linear Models):** Denote $(\mathbf{W}^\star)^\mathsf{T} = [\mathbf{w}^{\star(1)}, \ldots, \mathbf{w}^{\star(t)}] \in \mathbb{R}^{r \times t}$ and $\mathbf{B}^\star = [\mathbf{b}^{\star(1)}, \ldots, \mathbf{b}^{\star(t)}] \in \mathbb{R}^{d \times t}$; hence the matrix of optimal regressors is given by $\mathbf{U}^\star(\mathbf{W}^\star)^\mathsf{T} + \mathbf{B}^\star$. There are several technically novel steps that are required for recovering the model parameters in (1) that combine both the representation and neighborhood models. In the general setting, even the task-specific $r$-dimensional regression coefficients $\{\mathbf{w}^{\star(i)}\}_{i \in [t]}$ are unknown and therefore, we are faced with the additional challenge of learning $\{\mathbf{w}^{\star(i)}\}_{i \in [t]}$ jointly along with the shared representation matrix $\mathbf{U}^\star$ and the task-specific sparse parameter vectors $\{\mathbf{b}^{\star(i)}\}_{i \in [t]}$. Note that in the AM framework proposed in Thekumparampil et al. (2021), the authors consider only the representation model and not the neighborhood model (which is a simpler case of LRS setting). Similarly, in Netrapalli et al. (2014), the authors design an AM algorithm for the problem of reconstructing the low rank and sparse components of a matrix if the matrix is provided as an input. However, in our setting, we only observe linear measurements of the individual columns of the parameter matrix. Therefore, informally speaking, our analysis is faced with the key challenge of combining both sets of complementary techniques in Thekumparampil et al. (2021); Netrapalli et al. (2014). This leads to the analysis of several crucial steps in each iteration: 1) We track the incoherence of several intermediate matrices corresponding to the latest estimates $\mathbf{W}^{(\ell)}, \mathbf{U}^{(\ell)}$ of $\mathbf{W}^\star, \mathbf{U}^\star$. 2) We also track the $\mathsf{L}_{2,\infty}$ norm of the matrix $(\mathbf{I} - \mathbf{U}^\star(\mathbf{U}^\star)^\mathsf{T})\mathbf{U}^{(\ell)}$ to make progress on learning $\mathbf{B}^\star$. In particular, the second step is the most technically involved component of our analysis.

**Organization:** In Sec. 2, we introduce the general low rank+sparse (LRS) framework. In Sec. 2.1, we provide theoretical guarantees for the canonical linear model in the LRS framework and provide a differentially private version in Sec. 2.2. We analyze a special setting in Appendix B as warm-up (Rmk. 3) while detailed proofs are delegated to Appendix C, D. In Sec. 3 and Appendix A, we provide experimental results on synthetic and real datasets. In Appendix E, we discuss how to obtain a good initialization for our methods in the realizable setting via Method of Moments.

## 2 LRS FRAMEWORK FOR META-LEARNING/PERSONALIZATION

**Notations:** $[m]$ to denotes the set $\{1, 2, \ldots, m\}$. For a matrix $\mathbf{A}$, $\mathbf{A}_i$ denotes $i^{\text{th}}$ row of $\mathbf{A}$. For a vector $\mathbf{x}$, $x_i$ denotes $i^{\text{th}}$ element of $\mathbf{x}$. We sometimes use $\mathbf{x}_j$ to denote an indexed vector; in this case $x_{j,i}$ denotes the $i^{\text{th}}$ element of $\mathbf{x}_j$. $||\cdot||_2$ denotes euclidean norm of a vector and the operator norm of a matrix. $||\cdot||_\infty, ||\cdot||_0$ will denote the $\ell_\infty$ and $\ell_0$ norms of a vector respectively. $||\cdot||_{2,\infty}, ||\cdot||_\mathsf{F}$ will be used to denote the $\mathsf{L}_{2,\infty}$ and Frobenius norm of a matrix respectively. For a sparse vector $\mathbf{v} \in \mathbb{R}^d$, we define the support $\mathsf{supp}(\mathbf{v}) \subseteq [d]$ to be a set of indices such that $v_i \neq 0$ for all $i \in \mathsf{supp}(\mathbf{v})$ and $v_i = 0$ otherwise. We use $\mathbf{I}$ to denote the identity matrix. $\widetilde{O}(\cdot)$ notation subsumes logarithmic factors. $\mathbf{X}^{(i)} \in \mathbb{R}^{m \times d}$ denotes the matrix of covariates for the $i^{\text{th}}$ task such that $\mathbf{X}_j^{(i)} = (\mathbf{x}_j^{(i)})^\mathsf{T}$. Similarly, we write $\mathbf{y}^{(i)}, \mathbf{z}^{(i)} \in \mathbb{R}^m$ to denote the task-specific response vector and noise vector respectively.

Let $\mathbf{x} \in \mathbb{R}^d$ be the input point, and let $\widehat{y} = f(\mathbf{x}; \boldsymbol{\Theta})$ be the predicted label using a DNN $f$ with parameters $\boldsymbol{\Theta}$. Let there be $t$ tasks/domains. Then, the goal is to personalize $\boldsymbol{\Theta}$ to $\boldsymbol{\Theta}^{(i)}$ for each task $i \in [t]$ such that a) $\boldsymbol{\Theta}^{(i)}$ does not over-fit despite a small number of data-points labelled as task $i$, b) $\{\boldsymbol{\Theta}^{(i)}, 1 \leq i \leq t\}$ can be stored and inferred fast even for large $t$, say for $t \geq 1M$, c) the framework allows enough flexibility to ensure that different tasks/domains/users can *contribute* their data at different levels of privacy risks. Our method LRS attempts to address all the three requirements using a simple low-rank+sparse approach: we model each $\boldsymbol{\Theta}^{(i)}$ as $\boldsymbol{\Theta}^{(i)} := \mathcal{U} \cdot \mathbf{W}^{(i)} + \mathbf{B}^{(i)}$, where $\mathcal{U}\mathbf{W}^{(i)}$ denotes a linear operation on $\mathbf{W}^{(i)}$ in a $r$-dimensional basis specified by $\mathcal{U}$ (can be very large for e.g. standard BERT with $\sim 110M$ parameters), and $\mathbf{B}^{(i)}$ is a $k$-sparse matrix. That is, we represent $\boldsymbol{\Theta}^{(i)}$ as a combination of a small number of parameter matrices represented by $\mathcal{U}$, along with a sparse set of weights that can be fine-tuned arbitrarily for a given task.

Note that due to low-dimensional and sparse representation, LRS should require relatively small number of points per task. In the next section, we formally prove this claim for a simple linear setting with Gaussian data. Furthermore, for each task the additional number of parameters is relatively small – assuming $r$ and $k$ are small – which implies that memory cost can be controlled. The latency cost also remains same as the baseline model, assuming we can explicitly compute $\Theta^{(i)}$ on the fly. Finally, the model allows tasks/domains/users to contribute data to learn $\mathcal{U}$ with differential privacy (see Section 2.2), but it also admits domains/tasks who are not inclined to share data, and can just fine-tune their model privately by learning $\mathbf{W}^{(i)}$ and $\mathbf{B}^{(i)}$ in isolation.

**Comparison with Hu et al. (2021)**: LORA (Low Rank Adaptation of Large Language Models) was proposed by Hu et al. (2021) for meta-learning with large number of tasks at scale. Although the authors demonstrate promising experimental results, LORA only allows a central model (in a low dimensional manifold) and does not incorporate sparse fine-tuning. Hence, LORA becomes ineffective when the output dimension is small (say 1). The said limitation of LORA has been demonstrated in detailed experiments on the MovieLens 1M dataset (a standard recommendation dataset) in Sec. 3. Moreover, the low rank fine-tuning (as proposed in LORA) is limited when output dimension is small. Finally, LORA does not have any theoretical guarantees even in simple settings.

**Comparison with representation learning style models**: A recent line of work (Thekumparampil et al., 2021; Du et al., 2020; Tripuraneni et al., 2021; Boursier et al., 2022; Jain et al., 2021) proposes a similar model to LRS but is specific to representation learning. These papers only consider a low-rank representation of the task parameters but no sparse fine-tuning. Moreover, the proposed algorithms in these papers have not been explored at scale and mostly been applied to vanilla linear models (without considering privacy constraints or extension to more complex models). Even from a theoretical point of view, these methods do not apply in our case due to the additional non-convex sparsity constraint.

**Comparison with Prompt-based and Batch-norm Fine-tuning**: Another popular approach for personalization is to use prompt-based or batch-norm based fine-tuning (Wang et al., 2022; Liu et al., 2021; Lester et al., 2021); this usually involves a task-based feature embedding concatenated with the covariate. Note that in a linear model, such an approach will only lead to an additional scalar bias which can be easily modeled in our framework; thus our framework is richer and more expressive with a smaller number of parameters. We have compared against such techniques in our experiments and demonstrated their limitations in both real and synthetic datasets (see Sec. 3).

**Private Meta-learning:** Model-personalization is a key application of meta-learning, where we wish to have a personalized model for each user i.e. each *user* represents a task. Due to sensitivity of user-data, we would want to preserve privacy of each *user* for which we use user-level $(\epsilon, \delta)$-DP as the privacy notion (see Definition 1). In this setting, each user $i \in [t]$ holds a set of data samples $D^{(i)} = \{\mathbf{x}_j^{(i)}, 1 \le j \le m\}$. Furthermore, users interact via a central algorithm that maintains the common representation matrix $\mathcal{U}$ which is guaranteed to be DP w.r.t. all the data samples of any single user. The central algorithm publishes the current $\mathcal{U}$ to all the users (a.k.a. on a billboard) and obtains further updates from the users. It has been shown in prior works (Jain et al., 2021; Chien et al., 2021; Thakkar et al., 2019) that such a billboard mechanism allows for significantly more accurate privacy preserving methods while ensuring user-level privacy. In particular, it allows learning of $\mathcal{U}$ effectively, while each user can keep a part of the model which is personal to them, e.g., the $\mathbf{W}^{\star(i)}, \mathbf{B}^{\star(i)}$'s in our context. See (Jain et al., 2021, Section 3) for more details about billboard model in the personalization setting. Traditionally, such model of private computation is typically called the billboard model of DP, which in turn is a subclass of joint DP (Kearns et al., 2014).

**Definition 1.** *Differential Privacy Dwork et al. (2006b;a); Bun & Steinke (2016) A randomized algorithm $\mathcal{A}$ is $(\varepsilon, \delta)$-differentially private if for any pair of data sets $D$ and $D'$ that differ in one user (i.e., $|D \triangle D'| = 1$), and for all $S$ in the output range of $\mathcal{A}$, we have*

$$\Pr[\mathcal{A}(D) \in S] \le e^{\varepsilon} \cdot \Pr[\mathcal{A}(D') \in S] + \delta,$$

*where probability is over the randomness of $\mathcal{A}$. Similarly, an algorithm $\mathcal{A}$ is $\rho$-zero Concentrated DP (zCDP) if $D_{\alpha}(\mathcal{A}(D)||\mathcal{A}(D')) \le \alpha\rho$, where $D_{\alpha}$ is the Rényi divergence of order $\alpha$.*

In Definition 1, when we define the notion of neighborhood, we define it w.r.t. the addition (removal) of a single user (i.e., additional removal of all the data samples $D_i$ for any user $i \in [t]$). In the literature Dwork & Roth (2014), the definition is referred to as user-level DP.

### 2.1 LINEAR LRS: ALGORITHM AND ANALYSIS

In this section, we describe our LRS framework for the linear setting, provide an efficient algorithm for parameter estimation, and provide rigorous analysis under realizable setting with Gaussian data. We then extend our framework, algorithm and analysis to allow user-level differential privacy. Consider the linear LRS model introduced in Section 1 where we have $t$ $d$-dimensional linear regression tasks (indexed by $i \in [t]$) with $m$ samples $\{(\mathbf{x}_j^{(i)}, y_j^{(i)})\}_{j=1}^m$ being provided to each of them such that the $i^{\text{th}}$ sample for the $j^{\text{th}}$ task $(\mathbf{x}_j^{(i)}, y_j^{(i)})$ is generated independently according to eq. 1. So the problem reduces to that of designing statistically and computationally efficient algorithms to estimate the common representation learning parameter $\mathbf{U}^\star$ as well as task-specific parameters $\{\mathbf{w}^{\star(i)}\}_{i \in [t]}, \{\mathbf{b}^{\star(i)}\}_{i \in [t]}$. The ERM for this model assuming squared loss is given by:

$$\text{(LRS)} \qquad \text{minimize} \, \mathcal{L}(\mathbf{U}, \mathbf{W}, \mathbf{B}) = \sum_{i \in [t]} \sum_{j \in [m]} \frac{1}{2} \left( y_j^{(i)} - \langle \mathbf{x}_j^{(i)}, \mathbf{U}\mathbf{w}^{(i)} + \mathbf{b}^{(i)} \rangle \right)^2$$

$$\text{s.t. } \mathbf{U}^\mathsf{T}\mathbf{U} = \mathbf{I}, \, \left|\left|\mathbf{b}^{(i)}\right|\right|_0 \leq k \, \forall i \in [t] \text{ and } ||\mathbf{B}_i||_0 \leq \zeta \, \forall i \in [d], \qquad (2)$$

where $\mathbf{U} \in \mathbb{R}^{d \times r}$, $\mathbf{W} = [\mathbf{w}^{(1)} \, \mathbf{w}^{(2)} \, \dots \, \mathbf{w}^{(t)}]^\mathsf{T} \in \mathbb{R}^{t \times r}$ stores the task-specific coefficients, and $\mathbf{B} = [\mathbf{b}^{(1)} \, \mathbf{b}^{(2)} \, \dots \, \mathbf{b}^{(t)}] \in \mathbb{R}^{d \times t}$ stores the task-specific sparse vectors for *fine-tuning*. Note that LRS is non-convex due to: a) bilinearity of $\mathbf{U}, \mathbf{W}$, b) non-convexity of $\ell_0$ norm constraint.

We propose AMHT-LRS that handles the non-convexity in the objective and the constrained set by carefully combining alternating minimization for $\mathbf{U}, \mathbf{w}$ and $\mathbf{b}$ with hard thresholding to ensure sparsity of $\mathbf{b}$. Let $\mathsf{HT} : \mathbb{R}^d \times \mathbb{R} \to \mathbb{R}^d$ be a *hard thresholding* function that takes a vector $\mathbf{v} \in \mathbb{R}^d$ and a parameter $\Delta$ as input and returns a vector $\mathbf{v}' \in \mathbb{R}^d$ such that $v_i' = v_i$ if $|v_i| > \Delta$ and 0 otherwise. Let $\mathbf{U}^{+(\ell-1)}, \{\mathbf{w}^{(i,\ell-1)}\}_{i \in [t]}$ and $\{\mathbf{b}^{(i,\ell-1)}\}_{i \in [t]}$ be the latest iterates at the beginning of the $\ell^{\text{th}}$ iteration. First, for each task $i \in [t]$, given estimates $\mathbf{U}^{+(\ell-1)}, \mathbf{w}^{(i,\ell-1)}$, we can update $\mathbf{b}^{(i,\ell)}$ by solving the following problem:

$$\text{argmin}_{\mathbf{b} \in \mathbb{R}^d} \left|\left| \mathbf{X}^{(i)}(\mathbf{U}^{+(\ell-1)}\mathbf{w}^{(i,\ell-1)} + \mathbf{b}) - \mathbf{y}^{(i)} \right|\right|_2 \text{ such that } ||\mathbf{b}||_0 \leq k. \qquad (3)$$

While the problem is non-convex, we can still apply a projected gradient descent algorithm which reduces to iterative hard thresholding. In particular, we use Algorithm 2 for the $i^{\text{th}}$ task where in each iteration, we run a gradient descent step on the parameter vector estimate $\mathbf{b}$ (of $\mathbf{b}^{\star(i)}$) and subsequently apply $\mathsf{HT}(\cdot, \Delta)$ function where $\Delta > 0$ is set appropriately. Next, given estimates $\mathbf{U}^{+(\ell-1)}, \mathbf{b}^{(i,\ell)}$, we can update $\mathbf{w}^{(i,\ell)}$ by solving the following task-specific optimization problem

$$\text{argmin}_{\mathbf{w} \in \mathbb{R}^r} \left|\left| \mathbf{X}^{(i)}(\mathbf{U}^{+(\ell-1)}\mathbf{w} + \mathbf{b}^{(i,\ell)}) - \mathbf{y}^{(i)} \right|\right|_2 \text{ for each } i \in [t]. \qquad (4)$$

Subsequently, given the updated estimates of the task-specific parameters $\{\mathbf{w}^{(i,\ell)}\}_{i \in [t]}$ and $\{\mathbf{b}^{(i,\ell)}\}_{i \in [t]}$, we update $\mathbf{U}^{+(\ell)}$ (estimate of shared representation matrix) using:

$$\text{argmin}_{\mathbf{U} \in \mathbb{R}^{d \times r}} \sum_{i \in [t]} \left|\left| \mathbf{X}^{(i)}(\mathbf{U}\mathbf{w}^{(i,\ell)} + \mathbf{b}^{(i,\ell)}) - \mathbf{y}^{(i)} \right|\right|_2. \qquad (5)$$

followed by a QR decomposition of the solution. Note that the above two problems ((4) and (5)) can be solved using standard least squares regression methods. Finally, we must ensure independence of the estimates (which are random variables themselves) from the data that is used in a particular update. We can ensure such independence by using a fresh batch of samples in every iteration.

**Analysis:** As in prior works, we are interested in the few-shot learning regime when there are only a few samples per task. From information theoretic viewpoint, we expect the number of samples per task to scale linearly with the sparsity $k$ and rank $r$ and logarithmically with the dimension $d$. On the other hand, $\mathbf{U}^\star$ has $dr$ parameters and therefore, it is expected that the total number of samples across all tasks scales linearly with $dr$ which implies we would want the number of tasks $t$ to scale linearly with dimension $d$. Note that if the sparse vectors $\{\mathbf{b}^{\star(i)}\}_{i \in [t]}$ have the same support (or a high overlap between the supports), then the model parameters might not be uniquely identifiable. This is because, in that case, the matrix $\mathbf{B}^\star$ can be represented as a low-rank matrix. To establish identifiability of $\mathbf{U}^\star$ and sparse vectors $\mathbf{b}^\star$, we make the following assumption:

---

**Algorithm 1** AMHT-LRS

---

**Require:** Data $\{(\mathbf{x}_j^{(i)} \in \mathbb{R}^d, y_j^{(i)} \in \mathbb{R})\}_{j=1}^m$ for all $i \in [t]$, column sparsity $k$ of $\mathbf{B}$, $\left\|\Delta(\mathbf{U}^{+(0)}, \mathbf{U}^\star)\right\|_{\mathsf{F}} \leq \mathsf{B}$, $\max_i \|\mathbf{b}^{(i,0)} - \mathbf{b}^{\star(i)}\|_\infty \leq \gamma^{(0)}$, Parameter $\epsilon > 0$.

1: **for** $\ell = 1, 2, \ldots$ **do**
2:  Set $T^{(\ell)} = \Omega\left(\ell \log\left(\frac{\gamma^{(\ell-1)}}{\epsilon}\right)\right)$
3:  **for** $i = 1, 2, \ldots, t$ **do**
4:    $\mathbf{b}^{(i,\ell)} \leftarrow$ OptimizeSparseVector$((\mathbf{X}^{(i)}, \mathbf{y}^{(i)}), \mathbf{v} = \mathbf{U}^{+(\ell-1)}\mathbf{w}^{(i,\ell-1)}, \alpha = O\left(c_4^{\ell-1}\frac{\mathsf{B}}{\sqrt{k}}\right), \beta = O(c_5^{\ell-1}\mathsf{B}), \gamma = \gamma^{(\ell-1)}, \mathsf{T} = T^{(\ell)})$ for suitable constants $c_4, c_5 > 0$.
5:    $\mathbf{w}^{(i,\ell)} = \left((\mathbf{X}^{(i)}\mathbf{U}^{+(\ell-1)})^\mathsf{T}(\mathbf{X}^{(i)}\mathbf{U}^{+(\ell-1)})\right)^{-1}\left((\mathbf{X}^{(i)}\mathbf{U}^{+(\ell-1)})^\mathsf{T}(\mathbf{y}^{(i)} - \mathbf{X}^{(i)}\mathbf{b}^{(i,\ell)})\right)$
6:  **end for**
7:  Set $\mathbf{A} := \sum_{i \in [t]} \left(\mathbf{w}^{(i,\ell)}(\mathbf{w}^{(i,\ell)})^\mathsf{T} \otimes \left(\sum_{j=1}^m \mathbf{x}_j^{(i)}(\mathbf{x}_j^{(i)})^\mathsf{T}\right)\right)$ and $\mathbf{V} := \sum_{i \in [t]}(\mathbf{X}^{(i)})^\mathsf{T}\left(\mathbf{y}^{(i)} - \mathbf{b}^{(i,\ell)}\right)(\mathbf{w}^{(i,\ell)})^\mathsf{T}$. Compute $\mathbf{U}^{(\ell)} = \mathrm{vec}_{d\times r}^{-1}(\mathbf{A}^{-1}\mathrm{vec}(\mathbf{V}))$ and $\mathbf{U}^{+(\ell)} \leftarrow \mathsf{QR}(\mathbf{U}^{(\ell)})$
8:  $\gamma^{(\ell)} \leftarrow (c_3)^{\ell-1}\epsilon\mathsf{B}$ for a suitable constant $c_3 < 1$.
9: **end for**
10: Return $\mathbf{w}^{(\ell)}, \mathbf{U}^{+(\ell)}$ and $\{\mathbf{b}^{(i,\ell)}\}_{i \in [t]}$.

---

**Assumption 1** (A1). *Consider the matrix $\mathbf{B}^\star \in \mathbb{R}^{d\times t}$ whose $i^{\mathsf{th}}$ column is the vector $\mathbf{b}^{\star(i)}$. Then each row of $\mathbf{B}^\star$ is $\zeta$-sparse i.e. $\|\mathbf{B}_i^\star\|_0 \leq \zeta$ for all $i \in [d]$, and each column is $k$-sparse.*

Note that the orthonormal matrix $\mathbf{U}^\star$ cannot have extremely sparse columns otherwise it would be information theoretically impossible to separate columns of $\mathbf{U}^\star$ from $\mathbf{b}^\star$. Moreover, similar to Tripuraneni et al. (2021), we need to ensure that each task contributes to learning the underlying representation $\mathbf{U}^\star$. These properties can be ensured by the standard incoherence assumptions Tripuraneni et al. (2021); Collins et al. (2021); Netrapalli et al. (2014) and therefore, we have

**Assumption 2** (A2). *Let $\lambda_1^\star$ and $\lambda_r^\star$ be the largest and smallest eigenvalues of the task diversity matrix $(r/t)(\mathbf{W}^\star)^\mathsf{T}\mathbf{W}^\star \in \mathbb{R}^{r\times r}$. We assume that $\mathbf{W}^\star \in \mathbb{R}^{t\times r}$ and the representation matrix $\mathbf{U}^\star \in \mathbb{R}^{d\times r}$ are $\mu^\star$-incoherent i.e. $\|\mathbf{W}^\star\|_{2,\infty} \leq \sqrt{\mu^\star\lambda_r^\star}$ and $\|\mathbf{U}^\star\|_{2,\infty} \leq \sqrt{\frac{\mu^\star r}{d}}$.*

**Theorem 1.** *Consider the LRS problem equation 2 with $t$ linear regression tasks and samples obtained by equation 1. Let model parameters satisfy assumptions A1, A2. Let the row sparsity of $\mathbf{B}^\star$ satisfy $\zeta = O\left(t(r^2\mu^\star)^{-1}\sqrt{\frac{\lambda_r^\star}{\lambda_1^\star}}\right)$, and let $k = O\left(d \cdot (\frac{\lambda_r^\star}{\lambda_1^\star})^2\right)$. Suppose Algorithm 1 is initialized with $\mathbf{U}^{+(0)}$ such that $\left\|(\mathbf{I} - \mathbf{U}^\star(\mathbf{U}^\star)^\mathsf{T})\mathbf{U}^{+(0)}\right\|_{\mathsf{F}} = O\left(\sqrt{\frac{\lambda_r^\star}{\lambda_1^\star}}\right)$ and $\left\|\mathbf{U}^{+(0)}\right\|_{2,\infty} = O(\sqrt{\mu^\star r/d})$, and is run for $\mathsf{L} = \widetilde{O}(1)$ iterations. Then, with high probability, the outputs $\mathbf{U}^{+(\mathsf{L})}, \{\mathbf{b}^{(i,\mathsf{L})}\}_{i\in[t]}$ satisfy:*

$$\left\|(\mathbf{I} - \mathbf{U}^\star(\mathbf{U}^\star)^\mathsf{T})\mathbf{U}^{+(\mathsf{L})}\right\|_{\mathsf{F}} = \frac{\widetilde{O}(1)\sigma\mathsf{S}}{\sqrt{\mu^\star\lambda_r^\star}}, \left\|\mathbf{b}^{(i,\mathsf{L})} - \mathbf{b}^{\star(i)}\right\|_\infty \leq \frac{O(1)\sigma\mathsf{S}}{\sqrt{k}}, i \in [t], \tag{6}$$

*where $\mathsf{S} = \left(\mu^\star\sqrt{\frac{r^3 d}{mt}} + \sqrt{\frac{r^3}{m\lambda_r^\star}} + \sqrt{\frac{k}{m}}\right)$ provided the total number of samples satisfies:*

$$m = \widetilde{\Omega}\left(k + r^2\mu^\star\left(\frac{\lambda_1^\star}{\lambda_r^\star}\right)^2 + \frac{\sigma^2 r^3}{\lambda_r^\star}\right), mt = \widetilde{\Omega}\left(r^3 d\mu^\star\left(r(\mu^\star)^4(\lambda_r^\star)^2 k + \mu^\star\left(\frac{\lambda_1^\star}{\lambda_r^\star}\right)^2 + \sigma^2\left(1 + \frac{1}{\lambda_r^\star}\right)\right)\right)$$

*For a new task, modified AMHT-LRS (Alg. 6 in Appendix D) has the following generalization bound:*

$$\mathcal{L}(\mathbf{U}, \mathbf{w}, \mathbf{b}) - \mathcal{L}(\mathbf{U}^\star, \mathbf{w}^\star, \mathbf{b}^\star) = \widetilde{O}\left(\sigma^2\left(\mathsf{S}^2 + \frac{k+r}{m}\right)\right).$$

Note that the per-task sample complexity of our method roughly scales as $m = (r^3 + k)$, which is information theoretically optimal in $k$ and is roughly $r^2$ factor larger. Total sample complexity scales as $mt = kdr^4$, which is roughly $kr^3$ multiplicative factor larger than the information theoretic bound. Note that typically $r$ and $k$ are considered to be small, so the additional factors are small, but we

leave further investigation into obtaining tighter bounds for future work. Finally, the generalization error scales as $\sigma^2(r+k)/m$ which is nearly optimal. Note that, ignoring meta-learning, and directly optimizing the single-task error would lead to significantly larger error of $\sigma^2 d/m$.

**Remark 1** (Runtime and Memory). *The run-time of Algorithm 1 is dominated by the update for $\mathbf{U}^{+(\ell)}$. For each iteration $\ell$, Step 8 has a time complexity of $O((dr)^3 + (mt)(dr)^2)$; however in practice, a gradient descent step for the update of $\mathbf{U}^{(\ell)}$ can bring down the time complexity to $O(mtdr)$. Moreover, the memory usage of Algorithm 1 is $O((dr)^2 + tr^2)$.*

**Remark 2** (Initialization). *Note that Algorithm 1 has local convergence properties as described in Theorem 1. In practice, typically we use random initialization for $\mathbf{U}^{+(0)}$. However, similar to the representation learning framework in Tripuraneni et al. (2021), we can use the Method of Moments to obtain a good initialization. See Appendix E for more details.*

**Remark 3** (Special Settings). *In the setting where for each task, we just need to learn a single central model for all tasks and sparse fine-tune the weights for each task i.e. $\mathbf{w}^{\star(i)} = 1$ for all $i \in [t]$ is fixed, AMHT-LRS obtains global convergence guarantees (Theorem 4 in Appendix B). Moreover, if the central model $\mathbf{U}^\star$ is also frozen, then the task-based sparse fine-tuning reduces to standard compressed sensing. In the realizable setting, our framework recovers the standard generalization error of $\sigma^2 k/m$ in compressed sensing (Jain & Kar, 2017)[Chapter 7].*

**Remark 4** (Sample complexity comparison). *Note that the theoretical guarantees in representation models studied in Tripuraneni et al. (2021); Thekumparampil et al. (2021); Collins et al. (2021) cannot capture the sparse fine-tuning in each task (with potentially arbitrary magnitude). However, since our framework combines neighborhood and representation models, our sample complexity guarantees are sub-optimal by only a factor of $r^2$ (generalization error is optimal) when restricted to the special case of representation model (Thekumparampil et al., 2021).*

## 2.2 PRIVATE LINEAR LRS: PRIVACY PRESERVING META-LEARNING

In this section, we provide a user level DP variant of Algorithm 1 in the billboard model. We obtain DP for the computation of each $\mathbf{U}^{(\ell)}$ by perturbing the covariance matrix $\mathbf{A}$ and the linear term $\mathbf{V}$ in the algorithm with Gaussian noise to ensure that the contribution of any single user is protected. We start by introducing the function clip : $\mathbb{R} \times \mathbb{R} \to \mathbb{R}$ that takes as input a scalar $x$, parameter $\rho$ and returns $\mathsf{clip}(x, \rho) = x \cdot \min\left\{1, \frac{\rho}{x}\right\}$. We can extend the definition of clip to vectors and matrices by using $\mathsf{clip}(\mathbf{v}, \rho) = \mathbf{v} \cdot \min\left\{1, \frac{\rho}{\|\mathbf{v}\|_2}\right\}$ for a vector $\mathbf{v}$ and $\mathsf{clip}(\mathbf{A}, \rho) = \mathbf{A} \cdot \min\left\{1, \frac{\rho}{\|\mathbf{A}\|_\mathsf{F}}\right\}$ for a matrix $\mathbf{A}$. In order to ensure that

---

**Algorithm 2** OPTIMIZE SPARSE VECTOR

**Require:** Data $(\mathbf{X}, \mathbf{y}) \in \mathbb{R}^{m \times d} \times \mathbb{R}^m$ where we minimize $\|\mathbf{y} - \mathbf{X}(\mathbf{v}^\star + \mathbf{b}^\star)\|_2$ such that $\|\mathbf{b}^\star\|_0 \le k$. Estimate $\mathbf{v}$ (of $\mathbf{v}^\star$) and initialization $\mathbf{b}$ (of $\mathbf{b}^\star$). Iterations $\mathsf{T}$, parameters $\alpha, \beta, \gamma > 0$ and suitable constants $c > 0, 0 < c_1 < 1/2$.
1: **for** $j = 1, 2, \ldots, \mathsf{T}$ **do**
2: $\quad \mathbf{c} \leftarrow \mathbf{b} - \frac{1}{m} \cdot (\mathbf{X}^{(i)})^\mathsf{T}(\mathbf{X}^{(i)}\mathbf{b} + \mathbf{X}^{(i)}\mathbf{v} - \mathbf{y}^{(i)})$
3: $\quad \Delta \leftarrow \alpha + c_1\left(\gamma + \frac{\beta}{\sqrt{k}}\right)$ and $\mathbf{b} \leftarrow \mathsf{HT}(\mathbf{c}, \Delta)$
4: $\quad \gamma \leftarrow 2c_1\gamma + 2(\alpha + \frac{c_1}{\sqrt{k}}\beta)$
5: **end for**
6: Return vector $\mathbf{b}$.

---

Algorithm 1 is private, for input parameters $\mathsf{A}_1, \mathsf{A}_2, \mathsf{A}_3, \mathsf{A}_w$, we first clip the covariates and responses: for all $i \in [t], j \in [m]$, we will have $\widehat{\mathbf{x}_j^{(i)}} \leftarrow \mathsf{clip}\left(\mathbf{x}_j^{(i)}, \mathsf{A}_1\right)$, $\widehat{y_j^{(i)}} \leftarrow \mathsf{clip}\left(y_j^{(i)}, \mathsf{A}_2\right)$, $\widehat{(\mathbf{x}_j^{(i)})^\mathsf{T}\mathbf{b}^{(i,\ell)}} \leftarrow \mathsf{clip}\left((\mathbf{x}_j^{(i)})^\mathsf{T}\mathbf{b}^{(i,\ell)}, \mathsf{A}_3\right)$ and $\widehat{\mathbf{w}^{(i,\ell)}} \leftarrow \mathsf{clip}\left(\mathbf{w}^{(i,\ell)}, \mathsf{A}_w\right)$. Now, we can modify Line 7 in Algorithm 1 as follows (let $\mathsf{L}$ be the number of iterations of Alg. 1):

$$\mathbf{A} := \frac{1}{mt}\left(\sum_{i \in [t]}\left(\widehat{\mathbf{w}^{(i,\ell)}}(\widehat{\mathbf{w}^{(i,\ell)}})^\mathsf{T} \otimes \left(\sum_{j=1}^m \widehat{\mathbf{x}_j^{(i)}}(\widehat{\mathbf{x}_j^{(i)}})^\mathsf{T}\right)\right) + \mathbf{N}^{(1)}\right) \tag{7}$$

$$\mathbf{V} := \frac{1}{mt}\left(\sum_{i \in [t]}\sum_{j \in [m]} \widehat{\mathbf{x}_j^{(i)}}\left(\widehat{y_j^{(i)}} - \widehat{(\mathbf{x}_j^{(i)})^\mathsf{T}\mathbf{b}^{(i,\ell)}}\right)(\widehat{\mathbf{w}^{(i,\ell)}})^\mathsf{T} + \mathbf{N}^{(2)}\right) \tag{8}$$

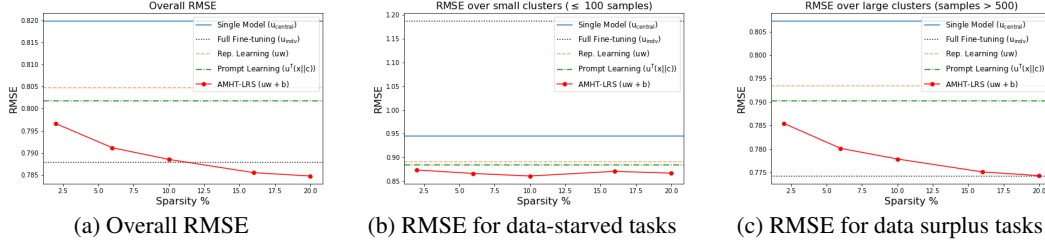

(a) Overall RMSE      (b) RMSE for data-starved tasks      (c) RMSE for data surplus tasks

Figure 1: Decrease in RMSE on MovieLens data for AMHT-LRS algorithm on increase in fine-tunable parameters. Note that AMHT-LRS outperforms other baselines for both data-starved and data-surplus tasks.

where, for some $\sigma_{\mathsf{DP}} > 0$, each entry of $\mathbf{N}^{(1)}$ is independently generated from $\mathcal{N}\left(0, m^2 \cdot \mathsf{A}_1^4 \cdot \mathsf{A}_w^4 \cdot \mathsf{L} \cdot \sigma_{\mathsf{DP}}^2\right)$; similarly, each entry of $\mathbf{N}^{(2)}$ is independently generated from $\mathcal{N}\left(0, m^2 \cdot \mathsf{A}_1^2(\mathsf{A}_2 + \mathsf{A}_3)^2\mathsf{A}_w^2 \cdot \mathsf{L} \cdot \sigma_{\mathsf{DP}}^2\right)$. We are now ready to state our main result:

**Theorem 2.** *Algorithm 1 (with modifications mentioned in equation 7 and equation 8) satisfies* $\sigma_{\mathsf{DP}}^{-2} - \mathsf{zCDP}$ *and correspondingly satisfies* $(\varepsilon, \delta)$-*differential privacy in the billboard model, when we set the noise multiplier* $\sigma_{\mathsf{DP}} \geq 2\varepsilon^{-1}\sqrt{(\log(1/\delta) + \varepsilon)}$. *Furthermore, if* $\varepsilon \leq \log(1/\delta)$, *then* $\sigma_{\mathsf{DP}} \geq \varepsilon^{-1}\sqrt{8\log(1/\delta)}$ *suffices to ensure* $(\varepsilon, \delta)$-*differential privacy.*

Next, we characterize the generalization properties of modified AMHT-LRS:

**Theorem 3.** *Consider the LRS problem equation 2 with all parameters* $m, t, \zeta$ *obeying the bounds stated in Theorem 1 and furthermore,* $t = \widetilde{\Omega}(\frac{(rd)^{3/2}\sqrt{\log(1/\delta)+\epsilon}}{\epsilon}\mu^\star)$. *Suppose we run AMHT-LRS (Step 7 in Alg. 1 replaced with 7 and 8) for* $\mathsf{L} = \widetilde{O}(1)$ *iterations with* $\mathsf{A}_1 = \widetilde{O}(\sqrt{d})$, $\mathsf{A}_2 = \widetilde{O}\left(\sqrt{\mu^\star\lambda_r^\star} + (\max_i \|\mathbf{b}^{\star(i)}\|_2)\right)$, $\mathsf{A}_3 = \widetilde{O}\left(\lambda_r^\star\sqrt{\frac{\mu^\star}{\lambda_1^\star}}\right)$, $\mathsf{A}_w = \widetilde{O}(\sqrt{\mu^\star\lambda_r^\star})$. *Then, with high probability, generalization error for a new task satisfies:*

$$\mathcal{L}(\mathbf{U}, \mathbf{w}, \mathbf{b}) - \mathcal{L}(\mathbf{U}^\star, \mathbf{w}^\star, \mathbf{b}^\star) = \widetilde{O}\left(\sigma^2\mathsf{S}^2 + \frac{dr^2(\log(1/\delta) + \epsilon)(\lambda_r^\star\mu^\star)^2}{\epsilon^2 t^2} \cdot (\kappa^2 + r^2 d^2)\right)$$

*where* $\mathsf{S} = \left(\mu^\star\sqrt{\frac{r^3 d}{mt}} + \sqrt{\frac{r^3}{m\lambda_r^\star}} + \sqrt{\frac{k}{m}}\right)$, $\eta = \widetilde{O}\left(t^{-1}\mu^\star r^2 d^{3/2}\left(1 + \sqrt{\frac{\lambda_r^\star}{\lambda_1^\star}} + \max_{i\in[t]}\frac{\|\mathbf{b}^{\star(i)}\|_2}{\sqrt{\mu^\star\lambda_r^\star}}\right)\sigma_{\mathsf{DP}}\right)$ *and* $\kappa = 1 + \sqrt{\frac{\lambda_r^\star}{\lambda_1^\star}} + \max_{i\in[t]}\frac{\|\mathbf{b}^{\star(i)}\|_2}{\sqrt{\mu^\star\lambda_r^\star}}$.

Note that the modified AMHT-LRS ensures $(\epsilon, \delta)-$differential privacy without any assumptions. However Thm. 3 still has good generalization properties; moreover, the per-task sample complexity guarantee $m$ still only needs to scale polylogarithmically with the dimension $d$. In other words, our algorithm can ensure good generalization along with privacy in data-starved settings as long as the number of tasks is large - scales as $\sim d^{3/2}/\epsilon$. Similarly, generalization error for a new task has two terms: the first has a standard dependence on noise $\sigma^2$ and the second has a scaling of $d^3(\epsilon t)^{-2}$ which is standard in private linear regression and private meta-learning (Smith et al., 2017; Jain et al., 2021). Detailed proofs of our main results namely Theorems 1,3 are delegated to Appendix C, D.

## 3 EMPIRICAL RESULTS

In this section, we conduct an empirical study of AMHT-LRS with the following two goals: a) demonstrate that personalization with AMHT-LRS indeed improves accuracy for tasks with a small number of points, b) for a fixed budget of parameters, AMHT-LRS is significantly more accurate than existing baselines. For simplicity, we fix the model class to be linear and consider the following baselines: 1) **Single Model** ($\mathsf{u}_{\mathsf{central}}$): learns a single model for all tasks, 2) **Full Fine-tuning** ($\mathsf{u}_{\mathsf{indv}}$) separate model for each task aka standard fine-tuning, 3) **Representation Learning or Rep. Learning** (uw): only low rank model (Thekumparampil et al., 2021) and 4) **Prompt Learning** $\mathbf{u}^\mathsf{T}(\mathbf{x} \| \mathbf{c})$: Modified covariate by concatenation with a task-embedding vector. Note that the models considered in Hu et al. (2021); Chua et al. (2021); Denevi et al. (2018) all reduce to Full fine-tuning models (that have a high memory footprint) in the experimental settings that we consider. Also, the

above approaches are not just restricted to linear models and can be extended to complex model classes such as Neural Networks (see Appendix A for extension to 3 layer Neural Net architectures).

We conduct experiments on two datasets: **a. Synthetic dataset**: here, for each task $i \in [t]$, we generate $m = 100$ samples $\{(\mathbf{x}_j^{(i)}, y_j^{(i)})\}_{j \in [m]}$ where $\mathbf{x}_j^{(i)} \sim \mathcal{N}(\mathbf{0}, \mathbf{I}_{d \times d})$, $y_j^{(i)} = \langle \mathbf{x}_j^{(i)}, \mathbf{u}^\star w^{\star(i)} + \mathbf{b}^{\star(i)} \rangle$. We select $d = 150$, set $k, \zeta$, the column and row sparsity level of $\{\mathbf{b}^{\star(i)}\}_{i \in [t]}$ to be 10 and 5, respectively. We sample $\mathbf{u}^\star$ uniformly from the unit sphere; non-zero elements of $\{\mathbf{b}^{\star(i)}\}_{i \in [t]}$ and $w^{\star(i)}$ are sampled i.i.d. from $\mathcal{N}(0, 1)$ with the indices of zeros selected randomly.

**b. MovieLens Data**: The MovieLens 1M dataset comprises of 1M ratings of 6K users for 4K movies. Each user is associated with some demographic data namely gender, age group, and occupation in the MovieLens dataset. We partition the users into 241 disjoint clusters where each cluster represents a unique combination of the demographic data. Each user group thus represents a "task" in the language of our paper. We partition the data into training and validation in the following way: for each task, we randomly choose 20% movies rated by at least one user from that task and put all ratings made by users from that task for the chosen movie into the validation set. The remaining ratings belong to the training set. Based on the ratings in the training set, we fit a matrix of rank 50 onto the ratings matrix and obtain a 50 dimensional embedding of each movie. Thus we ensure that there is no data leakage while creating the embeddings. For each task, the samples consist of (movie embedding, average rating) tuples; the response is the average rating of the movie given by users in that task. The number of samples per task varies from 22 to 3070 - clearly many clusters are data starved. We use the training data to learn the different models (with some hyper-parameter tuning) mentioned earlier and use them to predict the ratings in the validation data.

*Empirical Observations on Synthetic Data*: Figure 2 shows that not only having a single model can lead to poor performance, but a fully fine-tuned model per task can also be highly inaccurate as scarcity of data per task can leading to over-fitting. Finally, low-rank representation learning as well as prompt tuning based techniques do not perform well due to lack of modeling power. In contrast, our method is able to exactly recover the underlying parameters – as also predicted by Theorem 1 – and provides 5 orders of magnitude better RMSE.

*Empirical Observations on MovieLens*:

The overall average validation RMSE for AMHT-LRS and the different baselines that we consider is shown in Fig. 1a against percentage of fine-tunable parameters used by the model. With respect to the single model as reference, in the linear rank-1 case, the representation learning and the prompt learning based baselines have 1 and 50 additional parameters per task respectively; they are unable to personalize well. In contrast, with only $10\%(= 5)$ additional parameters per task, AMHT-LRS has smaller RMSE than fully fine-tuned model, which re-

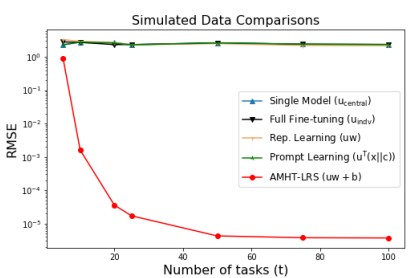

Figure 2: Decrease in RMSE on Synthetic data for AMHT-LRS on increase in fine-tunable parameters

quire 241x more parameters. However, for data-starved clusters/tasks (samples $< 100$), we observe that fully fine-tuning approach start to overfit. In contrast, our method outperforms other baselines for both data-starved and data-surplus tasks.

## 4 CONCLUSIONS

We presented a powerful theoretical framework to study meta-learning, and develop novel algorithms. In particular, our framework combines representation learning and neighborhood fine-tuning based approaches for meta-learning. We proposed AMHT-LRS method, that combines alternative mini-mization – popular in representation learning – with hard thresholding based methods. We rigorously proved that AMHT-LRS is statistically and computationally efficient, and is able to generalize to new tasks with only $O(r + k)$ samples, where $r$ is the representation learning dimension and $k$ is the number of fine-tuning weights. Finally, we extended our result to ensure that privacy of each *task* is preserved despite sharing information across tasks. Extending our framework to non-realizable setting and adversarial settings are critical future directions.

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
