# OpenReview forum: "Private and Efficient Meta-Learning with Low Rank and Sparse decomposition"
_ICLR.cc/2023/Conference — Submitted to ICLR 2023_

### Official Review · Reviewer_wf3D · 2022-10-25

**Confidence:** 3
**Correctness:** 3
**Technical Novelty And Significance:** 2
**Empirical Novelty And Significance:** 2
**Recommendation:** 5

**Clarity, Quality, Novelty And Reproducibility:**

Novelty: This paper is beyond [1] by adding a low-rank and a sparse assumption. It is something new in this area, but not very interesting.

[1] Differentially private model personalization. NeurIPS 2021

Quality: The theoretical analysis is technically sound.

Clarity: The paper is well-organized. However, the motivation for adding such a low-rank and a sparse assumption is unclear.

Reproducibility: The code needed to reproduce the experimental results is not provided.

**Strength And Weaknesses:**

Strengths:

1: The theoretical analysis is rigorous and solid.

2: The studied problem is interesting and could have a high impact on the community.

Weaknesses:

1: The motivation/intuition for adding such a low-rank and a sparse assumption is unclear.

2: Comparisons with [1] are not clear. What are the contributions/advantages beyond [1]?

[1] Differentially private model personalization. NeurIPS 2021

**Summary Of The Paper:**

This paper studies meta-learning approaches for representation learning in a linear setting with a low-rank and a sparse assumption. The authors proposed an alternating minimization method to solve the resulting problem. The sample complexity bound is also provided.

**Summary Of The Review:**

This paper studies meta-learning approaches for representation learning in a linear setting with a low-rank and a sparse assumption. The motivation for adding such a low-rank and a sparse assumption is unclear. Comparisons with the existing work [1] are not clear/enough.

---

> ### Author Response · Authors · 2022-11-16
> **Response to Review**
>
> ***Novelty: This paper is beyond [1] by adding a low-rank and a sparse assumption. It is something new in this area, but not very interesting.***
>
> In the Introduction of the paper, we have clearly mentioned that there are two existing approaches to meta-learning 1) Neighborhood models -  often involves sparse fine-tuning of a global model to specific tasks (see for example Guo et. al. 2020) 2) Representation Learning - involves learning a low-dimensional representation of points which can be used to train task-specific linear learners.  Please note that the paper that the reviewer has pointed out (Jain et. al. 2021) only studies the Representation Learning approach under privacy constraints and is thus a special case of our approach.
>
> The paper by Jain et. al. 2021 has studied a private optimization problem in the meta-learning set-up but that does not mean that the meta-learning problem has been completely solved. The motivation for combining the low rank and sparse models is to design algorithms for achieving the best of both worlds - Neighborhood  models and Representation learning. More importantly, our goal was to theoretically study the combination of the two aforementioned models in both private and non-private settings. While validating as well, our method AMHT-LRS outperforms Representation learning approach in both synthetic and real-world datasets.
>
> [1] Differentially private model personalization. NeurIPS 2021 (Jain et. al. 2021)
>
> ***Clarity: The paper is well-organized. However, the motivation for adding such a low-rank and a sparse assumption is unclear.***
>
> Please see the answer above. We have already provided detailed motivation for our problem set-up in the Introduction. We will be happy to clarify if the reviewer can point out ***more specifically what is unclear***.
> Studying theoretical guarantees of statistical recovery problems where the parameters allow a low rank+sparse decomposition is very well-motivated in model compression and has a rich literature in many other settings as well. Below, we provide a list of selected papers that the reviewer can look at:
> 1) Sparse and Low-Rank Matrix Decompositions by  Chandrasekaran et. al.
> 2) Non-convex robust PCA by Netrapalli et. al.
> 3) Low-rank and sparse matrix decomposition via the truncated nuclear norm and a sparse regularizer by Xue et. al.
> 4) Sparse Plus Low Rank Matrix Decomposition: A Discrete Optimization Approach by Bertsimas et. al.
>
> ***Comparisons with the existing work [1] are not clear/enough.***
> Jain et. al. (2021) has  studied the meta-learning problem under privacy constraints in the Representation Learning set-up (see answer to first question or the paper). We have provided a detailed comparison with representation learning style models in Page 4 of the paper (see paragraph titled "Comparison with representation learning style models:") . Additionally, in the revised version, we have updated the paragraph by citing Jain et. al. 2021.
>
> ***Low Reproducibility***- We promise to release all necessary code after the notification on the paper.

---

> ### Author Response · Authors · 2022-11-19
> **Thanks!**
>
> We thank the reviewer for the detailed review again. If the concerns of the reviewer are clarified by our rebuttal response, then can we respectfully request the reviewer to increase their score from 5?
>
> Thanks Authors

---

### Official Review · Reviewer_6iVb · 2022-10-27

**Confidence:** 3
**Correctness:** 3
**Technical Novelty And Significance:** 3
**Empirical Novelty And Significance:** 2
**Recommendation:** 5

**Clarity, Quality, Novelty And Reproducibility:**

Overall the paper is not clearly written. It contains many contents in 9 pages, resulting in the paper being dense and hard to read.

The paper provides theoretical results of high quality. In contrast, the empirical evaluation is not sufficient. First, the Movielens dataset is usually used for recommender systems. I'm not sure it is reasonable for meta-learning methods.  Second, Movielens is the only real dataset used in the experiments. In addition, only linear models are compared in the main results. Nonlinear models such as NNs would achieve better performance, and they should be compared. I found such experiments in Appendix A, where a baseline method (full fine-tuning) is better than the proposed one. It is explained that the memory usage of the baseline is high, but it was not empirically measured and less convincing.

Although the proposed model is simple, the derived algorithm has a theoretical guarantee and is novel.

The reproducibility is low (no code is provided).

**Strength And Weaknesses:**

Strengths
- Rigorous theoretical analysis is provided.

Weaknesses
- The paper is densely written and hard to follow.
- The model looks too simple to solve complex real-world data.
- The empirical evaluation is limited.

**Summary Of The Paper:**

This paper studies a linear meta-learning model comprising task-specific parameters: a low-rank weight and a sparse bias. The authors propose an alternating optimization algorithm for it. The generalization bound is also derived. The empirical performance is evaluated on Movielens 1M and synthetic datasets.

**Summary Of The Review:**

This paper proposes a simple linear model with a theoretically-guaranteed algorithm. However, the empirical evaluation is limited and whether the model is really applicable to real-world tasks remains mostly unknown.

---

> ### Author Response · Authors · 2022-11-16
> **Response to Review**
>
> ***The paper provides theoretical results of high quality. In contrast, the empirical evaluation is not sufficient. First, the Movielens dataset is usually used for recommender systems. I'm not sure it is reasonable for meta-learning methods.***
>
> Please see the common response to all reviewers. Note that recommendation datasets are ideal for capturing the challenges of personalization in low data/domain regime. Therefore, learning the preferences of users with small amounts of user-specific data  can be cast as a meta-learning problem tailored to data-starved settings (as has been specifically considered in our paper).
>
> ***Second, Movielens is the only real dataset used in the experiments. In addition, only linear models are compared in the main results. Nonlinear models such as NNs would achieve better performance, and they should be compared. I found such experiments in Appendix A, where a baseline method (full fine-tuning) is better than the proposed one. It is explained that the memory usage of the baseline is high, but it was not empirically measured and less convincing.***
>
> Again, we point to the common response to all reviewers for summary of the new experimental results that have been added to the revised manuscript (taking into account the comments of the reviewer)
>
> Regarding the Movielens dataset, we have now provided the detailed memory usage of the baselines (full fine-tuning in particular) and demonstrated the gains in memory footprint obtained by our method (See Table 1 in Page 18). In addition, we have provided similar experiments using Nonlinear models (Neural Networks) on two additional datasets - Netflix and Jester. In both these datasets, our proposed method AMHT-LRS outperforms the baselines (in particular full-finetuning and central models) by a significant margin (for both linear and non-linear models).
>
> ***Low Reproducibility***- We promise to release all necessary code after the notification on the paper.

---

> ### Author Response · Authors · 2022-11-19
> **Thanks!**
>
> We thank the reviewer for the detailed review again. If the concerns of the reviewer are clarified by the additional experiments provided in the revised version of the paper, then can we respectfully request the reviewer to increase their score from 5?
>
> Thanks
> Authors

---

### Official Review · Reviewer_aEd6 · 2022-11-02

**Confidence:** 2
**Correctness:** 3
**Technical Novelty And Significance:** 3
**Empirical Novelty And Significance:** 2
**Recommendation:** 6

**Clarity, Quality, Novelty And Reproducibility:**

Quality:
The authors studied extensively the theoretical framework proposed, and provided computational bounds. The experimental evaluation needs a bit more work.

Clarity:
The paper is well-written, and the general idea is clear. When delving deeper into the theoretical part, the message becomes a bit more messy, and the authors rely heavily on mathematical formulas rather than plain english to explain the assumptions and the theory.

Originality:
The paper idea of combining two common approaches of meta learning is new to the best of my knowledge. One related the work that would be beneficial for the paper is: _"SPARK: co-exploring model SPArsity and low-RanKness for compact neural networks"_. That paper explores model compression for a single task using low rank and sparse decomposition.

**Strength And Weaknesses:**

Strengths:
- the problem addressed in the paper is of direct real world implications, where companies have access to user data from multiple services, and need to learn a good model for all of them.
- the authors considered the privacy implications of their scheme and proposed a modification to solve it.
- the authors analyzed theoretically their approach, and provided computational bounds for their method.
- the results achieved by the proposed methods are promising and validate the intuition behind the method.

Weaknesses:
- the authors evaluate the proposed method, but do not indicate whether the employed method is differentially private or not.
- the authors do not provide actual values for the DP parameters for their proposed method.
- the authors test their method in one real-world dataset (MovieLens). While this step shows some validation of the results, I think more evaluation with real datasets is beneficial (e.g. meta-dataset)
- the authors do not provide runtime values for their proposed method.
- the evaluation lacks intuition about the choice of the model's parameters, e.g. low dimensional embedding size.
- I am slightly confused about the results. In the synthetic experiment, the proposed method is much better than all methods by a big margin. This however is not the case in the MovieLens dataset, where the proposed method achieves on average similar results to full fine-tuning. An intuition about the discrepancy source would be nice.

**Summary Of The Paper:**

When facing an ML problem with multiple tasks, and few datapoints per task, meta learning is used as a mean to provide good results simultaneously on all tasks. Typical approaches to meta learning include 1) fine-tuning a large model for each task at hand, and 2) learning a low dimensional representation to be shared by all tasks and used for predictions. The first approach provides good results in practice, however, it is not always feasible due to memory and computational constraints. In contrast, the second approach is efficient, however, provides lower performance.

The authors of this paper propose a new solution that aims to combine the goods of both worlds. In particular, the authors propose learning 1) a shared low dimensional representation, and 2) a task specific fine-tuning of this low dimensional representation To this end, the authors employ a technique based on low rank and sparse decomposition of the original large weights.

The authors then analyze theoretically the proposed method, called AMHT-LRS, in terms of computational bounds, when considering a Gaussian setup and a linear projection scheme. Furthermore, they propose a small tweak to the proposed method in order to satisfy differential privacy for each of the tasks.

The authors finally empirically verify their results in a toy setup, and on a real world dataset, showing the advantages of their proposed method.

**Summary Of The Review:**

The paper provides a new approach for meta learning that combines two existing approaches. The proposed approach is intuitive and makes sense, and the authors study their method theoretically. The authors also validate their method using a toy experiment and a literature dataset. The authors discuss theoretically the privacy implications of their method, however, it is unclear whether this aspect has been validated empirically.

---

> ### Author Response · Authors · 2022-11-16
> **Response to Review**
>
> We thank the reviewer for the detailed review. Below, we have provided our pointwise responses.
>
> ***the authors evaluate the proposed method, but do not indicate whether the employed method is differentially private or not.***
>
> In the earlier version of the paper, our experiments employed the non-private version of the algorithms. However, taking into account the suggestion of the reviewer, we have now added simulation results validating the private version of AMHT-LRS (see Appendix A.1 and Fig 3 for details in the revised version) and provided detailed comparison results with private and non-private versions of other baselines (similar to [1]).
>
> ***the authors do not provide actual values for the DP parameters for their proposed method.***
>
> Please see the above answer. We have now provided detailed simulation results validating the private version of AMHT-LRS in a synthetic dataset in the revised version.
>
> ***the authors test their method in one real-world dataset (MovieLens). While this step shows some validation of the results, I think more evaluation with real datasets is beneficial (e.g. meta-dataset)***
>
> Please see the common response provided to all reviewers. In the revised version, we have now provided detailed experimental results on two other datasets - Netflix and Jester further validating the theoretical guarantees of AMHT-LRS.
>
> ***the authors do not provide runtime values for their proposed method.***
>
> Remark 1 in Page 7 describes the runtime and memory of our proposed method AMHT-LRS. We will be happy to clarify any confusion.
>
> ***the evaluation lacks intuition about the choice of model's parameters, e.g. low dimensional embedding size.***
>
> This is a good question. Interestingly, we found that the ratings matrix corresponding to the groups of users and their corresponding ratings can be very well approximated by a low rank matrix. This is because the singular values of the matrix have a decaying tail. We chose the dimension to be 50 since the approximation error of fitting a rank 50 matrix onto the ratings matrix was found to be very small empirically.
>
> ***I am slightly confused about the results. In the synthetic experiment, the proposed method is much better than all methods by a big margin. This however is not the case in the MovieLens dataset, where the proposed method achieves on average similar results to full fine-tuning. An intuition about the discrepancy source would be nice.***
>
> Good question again. Note that in the synthetic experiment, we kept the number of samples/task to be very small (in particular m=100) while the number of model parameters d was 150. Therefore full fine-tuning led to heavy overfitting. However, in the MovieLens dataset 1) the cluster sizes vary significantly - therefore on average, full fine-tuning does quite well 2) the effect of overfitting is not significant since noise is low. ***To test this hypothesis, we experimented with the MovieLens dataset where we added noise to the ratings - indeed, in that case the difference in performance between our proposed method and AMHT-LRS becomes significantly more pronounced. If the reviewer suggests, we can add this experiment to the paper as well.***
>
> Also, please note the data generation process for the synthetic experiment. In essence, we are doing parameter recovery there for the ground truth model hypothesis. The experiment emphasizes the fact that other baselines/approaches fail utterly in capturing this model class while LRS is a generalized approach and other models are only special cases of it.
>
> ***the authors discuss theoretically the privacy implications of their method, however, it is unclear whether this aspect has been validated empirically.***
>
> Please see the answer to the first question. We have now provided detailed simulation results validating the private version of AMHT-LRS in the revised version.
>
> [1] Differentially private model personalization. NeurIPS 2021
>
> ***Originality: The paper idea of combining two common approaches of meta learning is new to the best of my knowledge. One related the work that would be beneficial for the paper is: "SPARK: co-exploring model SPArsity and low-RanKness for compact neural networks". That paper explores model compression for a single task using low rank and sparse decomposition.***
>
> We thank the reviewer for pointing out this related paper. It seems that the SPARK paper is neither published nor present in Arxiv. In that regard, we are unsure if we can cite the paper or present a valid comparison with it. Please let us know if we are mistaken.

---

### Author Response · Authors · 2022-11-16
**Common Response to All Reviewers**

We thank the reviewers for their time and effort. We are grateful that the reviewers appreciate the extensive theoretical contributions of the paper. Taking into account the reviewer’s comments, we have now updated the paper substantially with several new experiments on real-world datasets. We summarize them below (please look at Appendix A for details):

1)  We have now added experiments on the Netflix Challenge dataset. Similar to the MovieLens dataset already presented in the paper, we have first compared AMHT-LRS with other linear baselines (Single Model, Full fine-tuning, Rep. Learning, Prompt learning). Next, we have extended AMHT-LRS for  non-linear models (neural networks) and compared them with the respective baselines (extended for the same non-linear model).

2) We have conducted the same experiments as above (linear and nonlinear) for the Jester dataset.

3) We have added simulation experiments to validate the privacy implications of our method AMHT-LRS similar to [1]. Here, in a synthetic dataset, we validated the performance of the private version of AMHT-LRS and compared it with appropriate baselines.

Some reviewers have questioned the choice of recommendation datasets for our experiments. Please note that regarding practice,  the applications we envisage involve  personalization/meta-learning in large-scale regime (***see Line 1 in Introduction***) where the number of users/domains is large but the number of ratings per user is very small implying that the number of data-points per domain is  small (much smaller than the number of model parameters). We believe that recommendation datasets such as Jester, Movielens, Netflix perfectly capture this sort of scenario and therefore validate our theoretical guarantees tailored to such settings. On the other hand, current meta-datasets do not capture such challenges and are not so interesting. Hence our real-world experiments are conducted on recommendation datasets to understand the extent of personalization that is possible via our method in data-starved domains.

Finally, we also urge the reviewers to consider that our primary contributions are theoretical in nature and experiments are provided for validation of theoretical guarantees.

[1] Differentially private model personalization. NeurIPS 2021

---

### Author Response · Authors · 2022-11-16
**Approaching deadline for end of Discussion Phase 1**

We thank the reviewers for all their time and effort in handling the paper. Please note that we have submitted a detailed rebuttal addressing all questions raised by the reviewers.

***We apologize for submission of the rebuttal so late into the Discussion Phase 1***. However, it will be incredibly helpful if it is still possible to let us know if the questions raised have been answered appropriately and if we can clarify anything else. We will do our best to do so before the deadline.

Thanks again.

---

### Decision · Program_Chairs · 2023-01-20

**Decision:**

Reject

**Justification For Why Not Higher Score:**

The theoretical proofs in the paper are long, and while there is some experimental work presented, the paper may be better suited to a more (explicitly) theory-oriented venue.

**Justification For Why Not Lower Score:**

NA

**Metareview: Summary, Strengths And Weaknesses:**

The paper is motivated by a question in meta-learning. Ultimately this is a very theoretical paper that boils down to estimating the sum of a rank-r and a k-column sparse matrix. The question itself is neat, and even well-motivated. The entire proof of correctness is rather long. The authors efforts in responding to the reviewers, including providing additional experiments is greatly appreciated. Ultimately, I'm not convinced that there would be sufficiently wide interest in this work at ICLR, the results will be better appreciated in venues such as COLT or ALT, and the authors should consider submitting to those conferences.